# Emotion recognition while applying cosmetic cream using deep learning from EEG data; cross-subject analysis

Jieun Kim[1,2]*, Dong-Uk Hwang[1], Edwin J. Son[1], Sang Hoon Oh[1], Whansun Kim[1], Youngkyung Kim[3], Gusang Kwon[3]

1 Division of Fundamental Research on Public Agenda, National Institute for Mathematical Sciences, Daejeon, South Korea, 2 AIRISS AI Team, Yuseong-gu, Deajeon, South Korea, 3 AMOREPACIFIC R&D Center, Yongin-si, Gyeonggi-do, South Korea

* jsje1102@nims.re.kr

**Data Availability Statement:** All files are available from the GitHub database(URL: https://github.com/jsje1102/AMORE_cream_emotion).

## Abstract

We report a deep learning-based emotion recognition method using EEG data collected while applying cosmetic creams. Four creams with different textures were randomly applied, and they were divided into two classes, "like (positive)" and "dislike (negative)", according to the preference score given by the subject. We extracted frequency features using well-known frequency bands, i.e., alpha, beta and low and high gamma bands, and then we created a matrix including frequency and spatial information of the EEG data. We developed seven CNN-based models: (1) inception-like CNN with four-band merged input, (2) stacked CNN with four-band merged input, (3) stacked CNN with four-band parallel input, and stacked CNN with single-band input of (4) alpha, (5) beta, (6) low gamma, and (7) high gamma. The models were evaluated by the Leave-One-Subject-Out Cross-Validation method. In like/dislike two-class classification, the average accuracies of all subjects were 73.2%, 75.4%, 73.9%, 68.8%, 68.0%, 70.7%, and 69.7%, respectively. We found that the classification performance is higher when using multi-band features than when using single-band feature. This is the first study to apply a CNN-based deep learning method based on EEG data to evaluate preference for cosmetic creams.

## 1. Introduction

Cosmetics are a very important consumer product for emotional satisfaction as well as skincare. Several studies have demonstrated that cosmetics have a positive effect on well-being and self-esteem [1–5]. Most of the consumer satisfaction evaluations for cosmetics have been conducted through questionnaires and interviews [3–7]. However, these measurements may not be direct because real-time measurements of an individual's feelings and oral or written comments may be distorted during the response process. For this reason, more objective methods based on psychophysiological or neurological measurements are required. Recently, Electroencephalography (EEG) feature extraction using deep learning in the fields of emotion and BCI (Brain-Computer Interface) have been conducted, and it is showing good performance [8–14].

**Funding:** This research was supported by the National Institute for Mathematical Sciences (NIMS) grant funded by the Korean government (No. NIMS-B21910000, No. NIMS-B22720000), awarded to JK, DH, ES, SHO, and WK. Also this research was supported by a grant of the Korea Health Technology R&D Project through the Korea Health Industry Development Institute (KHIDI), funded by the Ministry of Health & Welfare, Republic of Korea (grant number: HP20C0083), awarded to YK and GK.

**Competing interests:** The authors have declared that no competing interests exist.

Electroencephalography (EEG) is a brain activity recording method. Among brain imaging technologies, EEG can be measured with the highest temporal resolution (less than 1s). EEG can non-invasively measure the electrical activity generated by the synchronized activity of tens to hundreds of thousands of neurons using electrodes placed on the scalp. In general, EEG has a specific frequency band during spontaneous firing or specific activity. Neural oscillation or brain wave is a rhythmic or repetitive pattern of neural activity in the central nervous system. The neuroscience community calls these specific frequency bands delta (1–3 Hz), theta (4–7 Hz), alpha (8–12 Hz), beta(13–38 Hz), and gamma (39–42 Hz) to distinguish them. The use of EEG for emotion recognition has a long history. To measure emotion with EEG, Russell's two-dimensional model has been used in many studies [15–18]. Human emotions can be conceptualized within Russell's 2-D (Valance (pleasure)-Arousal) or 3-D model (Pleasure-Arousal-Dominance) [19,20]. Valance indicates the degree to which an emotion is positive (like) or negative (dislike), whereas arousal indicates its strength, that is, the strength of the related emotional state [19,21,22]. According to recent papers published from 2016 to 2019, common approaches to stimulating the user's emotions are music, music videos, photos, video clips, and virtual reality [23]. There have also been many studies on the classification of emotions towards fragrance [24–29]. However, there have been few studies on emotions induced by touch compared to other sensory stimuli. Several studies have used EEG data analysis to investigate neural responses to emotional touch using a device that gently strokes a fabric placed on a subject's forearm [30–32]. They found inhibition of alpha rhythms in the somatosensory cortex of the contralateral hemisphere, reflecting aspects of basic tactile processing. Increased beta bands in the parietal-frontal region and gamma oscillations in the somatosensory cortex and frontal regions were found to be associated with a pleasant fabric feel. In another study examining interpersonal contact, beta oscillations in the middle prefrontal cortex were found to correlate with intimate interpersonal contact (hugging, massage, and rubbing) [33]. In a study evaluating the haptic preference of a washing machine handle, valence values were found to have a high correlation with the gamma band in the central frontal lobe [34]. Raheel et al. recognized human emotions using physiological signals observed in response to tactile-enhanced multimedia content, including three human senses (tactile, visual, and auditory). When viewing video clips, a fan and heater were used to induce the tactile sensation of hot and cold air [35]. The fusion of EEG, photoplethysmography (PPG), and galvanic skin response (GSR) yielded a classification accuracy of 79.76% when interacting with tactile-enhanced multimedia. As a result of a search for emotion recognition research using EEG during the use of cosmetics, only 3 papers were found. EEG and GSR were used to measure the emotions that occur when lip balm is applied to the lips [36]. They have demonstrated that one emollient can trigger different emotional responses while another lip balm is applied to the lips. According to Balconi et al., the EEG characteristics of the consumer's cosmetic touch were found during real-time store searches [37]. An increase in the right frontal beta band of the cerebrum has been shown, emphasizing the importance of touch in the consumer experience. Recently, Gabriel developed real-time right- or left-moving particles with a digital artist to measure and visually express consumers' emotions in real-time while applying two different creams [38]. They conducted an analysis based on previous finding that decreased activity of the alpha band in the left/right frontal lobe was associated with positive/negative emotions.

## 1–1 Traditional machine learning approach for emotion recognition

Extensive research has been conducted to identify emotional states (mainly visual or odor-induced emotions) by EEG signals using traditional machine learning. In machine learning approaches, features must be extracted manually and fed to train a classifier. In recent years,

various feature extraction methods have been proposed [39]. Most commonly, EEG features are extracted in the frequency domain. The EEG signal can be divided into five frequency bands: delta (1–3 Hz), theta (4–7 Hz), alpha (8–12 Hz), beta (13–38 Hz), and gamma (39–42 Hz) Frequency domain feature extraction mainly uses the Fourier and wavelet transforms [40]. Koelstra et al. extracted power spectral characteristics from all 32 electrodes to achieve an emotion classification accuracy of 62.0% in two-class arousal classification and 57.6% in two-class valance using the DEAP dataset [41]. Candra et al. extracted the wavelet entropy characteristics from alpha, beta, and gamma bands and classified emotions using a support vector machine (SVM) [42]. They also compared the results using EEG segments of 10 different window sizes and found that a window of 3 to 12 s was best for emotion recognition. In the time domain, EEG feature extraction methods use statistical analysis (mean, standard deviation, first and second derivative), Hjorth features, fractal dimensions, non-stationary indices, and higher-order crossings.

According to Yin et al., both the frequency and time domains were considered to find the EEG features related to emotion [43]. They extracted 160 features for the power values of the five bands of each channel and 56 features for the power differences in the left and right hemispheres. Furthermore, for the time domain features, 224 features corresponding to the mean, variance, Shannon entropy, spectral entropy, kurtosis, and skewness for each channel were extracted. Thus, a total of 440 EEG features were extracted. They proposed a transfer recursive feature elimination (T-RFE) approach, which determines the optimal feature subset for cross-subject emotion classification, and rather than using all features, the optimal feature subset achieved better results. Chen and Zhang compared two different feature extraction approaches with four different machine learning classifiers and found that nonlinear dynamic features lead to higher accuracy [44]. While the above traditional machine learning-based methods are effective in classifying emotional states, these approaches require researchers to spend significant time and effort finding and designing various emotion-related features in the data.

## 1–2. Emotion recognition using EEG based on deep learning

In recent years, deep learning has attracted significant attention, with great success in various fields [45]. Deep learning consists of multiple processing layers to learn representations of data at multiple levels of abstraction. These methods have greatly improved the state-of-the-art in visual object recognition, speech recognition, and many other areas, such as genomics, and disease diagnosis. Deep learning automatically extracts meaningful features from raw data, making it possible to replace the hand-crafted features in conventional machine learning. It has become possible to automatically extract frequency, spatial, and time-domain features. Recent studies have used deep learning techniques for emotion recognition [Table 1]. However, despite the automatic feature extraction of deep learning, in most cases, raw data are not used, and specific features are extracted from EEG (mainly through time-frequency analysis) and then applied to deep learning [8,9,46–49]. The papers in Table 1 that used raw data were published by Wang et al. [50] and Cimtay et al. [51], and the remainder used feature input-based deep learning.

Most of the previous EEG feature extraction methods focused only on the time and frequency dimensions, but recent research combined with spatial dimension information have been published. A method using a graph neural network, which is a method of viewing correlation between channels using spatial information, has been introduced [8,9]. Bashivan et al. proposed a deep recurrent-convolutional neural network (CNN) for EEG representation [10]. This architecture is designed to preserve the spatial, spectral, and temporal structures of the EEG. They fed the model a multispectral image that preserves the EEG topology. They used a

Table 1. Representative EEG-based emotion recognition studies using deep features.

| Papers | Input Feature | Deep learning architecture | Dataset | Evaluation method | Accuracy (%) for Valance (two classes) |
|---|---|---|---|---|---|
| Chao et al. [46] | Multiband feature matrix | Capsule network | DEAP* | 10-fold cross-validation | 66.73 |
| Zhong et al. [8] | Frequency bands | Regularized graph neural network | SEED** | Subject-dependent | 92.24 |
| | | | | Subject-independent | 85.3 |
| | | | | (Leave-one-subject-out) | |
| Tripathi et al. [47] | Statistical value | DNN + 2D CNN | DEAP | Leave-one-subject-out | 81.4 |
| Wen et al. [52] | Pearson feature | 2D CNN | DEAP | Subject-dependent | 77.98 |
| Wang et al. [50] | Raw data | 3-D CNN | DEAP | - | 72.1 |
| Yin et al. [9] | Differential entropy | Fusion model of LSTM and Graph CNN | DEAP | Subject-dependent | 90.45 |
| | | | | Subject-independent | 84.81 |
| | | | | (5 fold cross-validation) | |
| Jin et al. [48] | Channel-wise feature | LSTM | DEAP | 10-fold cross-validation | 98.93 |
| | | | SEED | | 99.63 |
| Cimtay et al. [51] | Raw data | InceptionResnetV2 + GAP+dense layer (4 layers) | DEAP | Leave-one-subject-out | 72.81 |
| | | | SEED | | 86.56 |
| | | | LUMED*** | | 81.8 |
| Tang et al. [53] | Differential Entropy from 32 channels, 5 bands | Bimodal deep denoising auto encoder + Bimodal-LSTM | SEED | 10-fold | 93.97 |
| | | | DEAP | cross-validation | 83.53 |
| Zheng et al. [49] | Frequency bands and channels | DBN | SEED | Subject-dependent | 86.08 |
| Pandey et al. [54] | Variational mode decomposition (VMD) | DNN | DEAP | Leave-two-subject-out | 62.5 |

DEAP*: A public dataset measuring emotion (valance, liking, arousal, and dominance) with 8 physiological signals and EEG signals from 32 subjects for music video stimuli., SEED**: A dataset measuring emotion (positive, neutral, and negative) by EEG of 15 subjects to Chinese movie clip stimuli. LUMED***: A dataset measuring emotion (positive, neutral, andnegative) by multimodal measurement (include visual data, peripheral physiological signals, and EEG) of 11 subjects for audio-visual stimuli. All datasets are open to public access.

2D CNN for each EEG frame to extract spatial features and a combination of long short-term memory (LSTM) and 1D CNNs to model temporal features. They also visualized the output of the 2D CNN to observe important bands and regions of the human brain. The proposed model was tested using a working memory dataset recorded when instructing participants to memorize various numbers of letter. Li et al. proposed a preprocessing method that transforms multi-channel EEG data into 2D frame representations and integrates CNNs and RNNs into cognitive-emotional states [55]. They extracted power spectral density (PSD) from different EEG channels and mapped them to a 2D plane to construct an EEG multidimensional feature image (MFI), and then they adopted a CNN to learn temporal image patterns from the EEG MFI sequences, while LSTM was used to classify human emotions. Chao et al. [56] trained a multiband feature matrix (MFM) using the frequency domain and spatial features of the EEG channel as an input image in the capsule network (CapsNet). They found that the three features (valence, arousal, and dominance) included in the MFM are complementary and that the capsule network is suitable for mining and utilizing the three correlation features. Wang et al. developed an EmotioNet model using 3D tensors (2D electrode topological structure × time samples) as input images [50]. They classified valance with 72.1% accuracy on raw EEG data.

Phan et al. proposed a method to express spatial features using multi-scale kernel-based CNNs [57]. They aimed to describe the channel and frequency band correlations by selecting EEG features and transforming the extracted features into a 3D matrix. A multi-scale kernel CNN was constructed to determine the correlation differences between pairs of near and remote channels and to capture spatial information in short- and long-range dependencies. The proposed network not only learns the emotion patterns of 32 EEG signal channels but also considers the interactions between 4 frequency bands.

So far, CNN-based models of various forms have been developed for EEG analysis [9,47,50–52,57,58]. We used four merged frequency bands considering EEG channel position for CNN input data. We propose the following CNN models for the merged four-band input: 1) general stacked layer CNN 2) inception-like CNN that can express local and global relationships between EEG channels with various kernel sizes, 3) a parallel input CNN that separates and inputs each frequency band. For comparison, four single-band input CNN were compared with the case of using multi-band features.

One of the challenges in modeling emotional states from EEG data is finding features that do not change between subjects and within-subjects. In other words, it is important to find a general-purpose or subject-independent classification model because, for practical application completely independent subjects who do not participate in training are targeted. Subject-independent means that training was conducted on one subject group and testing was conducted on an entirely new subject group. In most of studies using DNN for emotion classification so far, there were cases where the data of the test subjects were included in training [9]. Subject-independent studies were found [8,51,54]. In [51], they completely separated the training, test, and validation sets for the DEAP, SEED, and LUMED datasets, respectively. They evaluated the model using the leave-one-subject-out cross-validation (LOSO CV) method. During training, if the accuracy of the validation set did not improve for 6 consecutive epochs, training was stopped and the test data were applied to the final model. In general, validation is used for the purpose of early stopping and preventing model overfitting during network training. However, most EEG studies did not address the overfitting problem. Pandey et al. used two subjects as test data and the remainder as training data in the DEAP dataset [54]. There was no validation in the Pandey et al.'s study. In their paper, they did not mention where the stopping point was during training in a DNN. That is, there is a concern about overfitting. Zong et al. reported that when creating a regularized graph neural network, concerns about overfitting were reduced by using a simple model [8]. However, the definition of a simple model is ambiguous, and it is not clear whether overfitting has occurred. We used the following method for more reliable overfitting prevention. When training up to maximum epochs, we saved the training model for every epoch and computed validation accuracy and loss. After that, two models were saved after finding the maximum validation accuracy and minimum validation loss, respectively. For each model, two confusion matrices were obtained from the test subject data.

## 1–3. Motivation of our study

Until now, most emotions have been classified for stimuli related to sight, hearing, and smell. There have been only a few attempts to measure consumers' feelings objectively while applying cosmetic creams. This study aimed to understand consumers' emotions with EEG while they apply creams. We created a general classification model for the classification of consumers' emotions during cosmetic cream use. In this study, meaningful results were obtained by deep learning for the emotion of touch induced while using the cream on 19 subjects.

Our study has the following characteristics:

1. LOSO CV was applied to identify common EEG patterns corresponding to emotion, independent of the subject.

2. Our model is less concerned about overfitting because we used a validation dataset to prevent it.

3. To effectively extract spatial features of EEG, EEG 2D layout images were used so that each channel position is preserved.

4. A merged band input image was created considering the correlation between multiple frequency bands.

5. The correlation between the channel and frequency band can be learned by using the feature in which the EEG 2D channel and frequency band are merged.

6. We compared the seven CNN models we proposed and found the best CNN structure.

This is one of the first studies on emotion recognition for creams using the latest neural network model with EEG data. Compared with previous studies, our study verified the model with a more thorough method (using LOSO CV, preventing overfitting by using validation dataset) in applying the deep learning method. In addition, a new attempt was made to recognize human emotions by presenting new stimuli, i.e., cosmetic creams. Compared with the research results presented so far, our study obtained comparable results using the developed deep learning method in emotion analysis, although the stimulus and dataset were different from ours.

## 2. Data acquisition

### 2–1. Subjects

In this study, 19 healthy Korean female subjects without brain disorders (mean age 34.1 years, range 24–45 years) participated. Participants were asked to avoid eating and drinking and to get adequate sleep 6 hours before the start of the study. This study was approved by the Amorepacific R&D Institutional Review Board (IRB: 2021-1CR-N38S). The purpose of the study and relevant information were provided to the participants. In addition, informed written consent was obtained before the study, and all experiments are performed under approved guidelines and regulations.

### 2–2. EEG measurement

In this study, EEG was measured using a dry Quick-20 wireless 20-channel system device (Cognionics Inc., San Diego, CA). EEG data is available in 19 channels: Fp1, Fp2, F7, F3, Fz, F4, F8, FC3, FCz, FC4, T3, C3, Cz, C4, T4, CP3, CPz, CP4, T5, P3, Pz, P4, T6, O1, Oz, and O2, and the data were referenced in A1 (left earlobe). The skin-electrode impedance was maintained at <2500 kOhm. The sampling rate was 500Hz, and images were saved in bdf format.

### 2–3. Experimental procedure

Participants underwent the experiment in a controlled room with an ambient temperature of 22°C and humidity of 50%. A comfortable sitting state was maintained and EEG equipment was worn (within 10 min). Fig 1 shows the appearance of a subject wearing an EEG device and participating in an experiment. The areas where the cream is to be applied were marked (circled) on the left forearm of each subject. Each experiment was conducted twice for a total of four types of cream, as shown in Fig 2. After measuring the resting state for 10 s with eyes

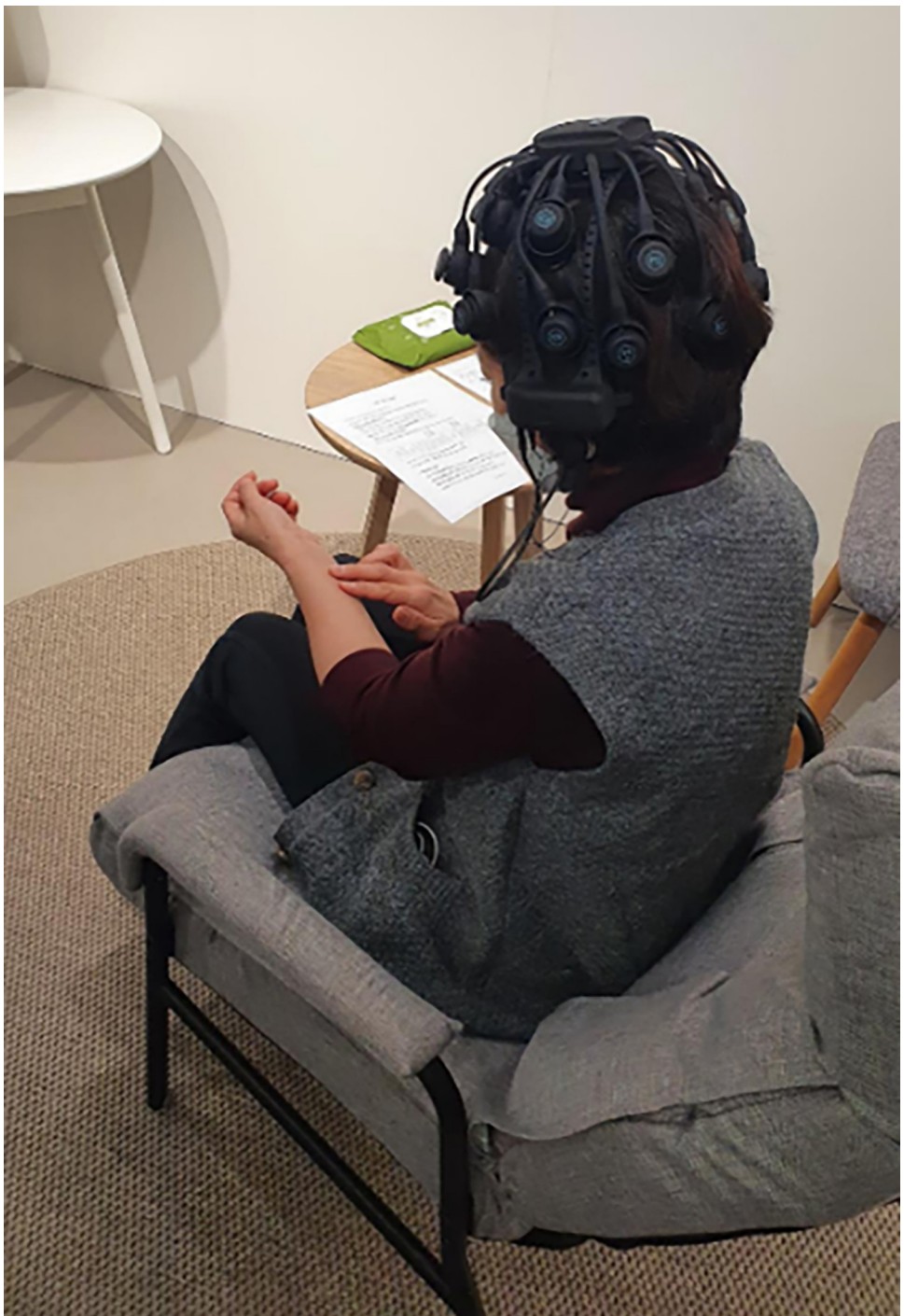

**Fig 1. A subject wearing an EEG device and participating in an experiment.**

open, a beep sound was played, and a certain amount of cream was applied to the forearm. For 30 s, the subject applied the cream by rubbing or tapping the cream within the marked circle using the index and middle fingers. Subjects were asked to focus on their emotions while applying the cream. When the beep was heard again, the subject stopped rubbing, lowered their arms comfortably, and concentrated on emotions for 20 s. When the last (third) beep

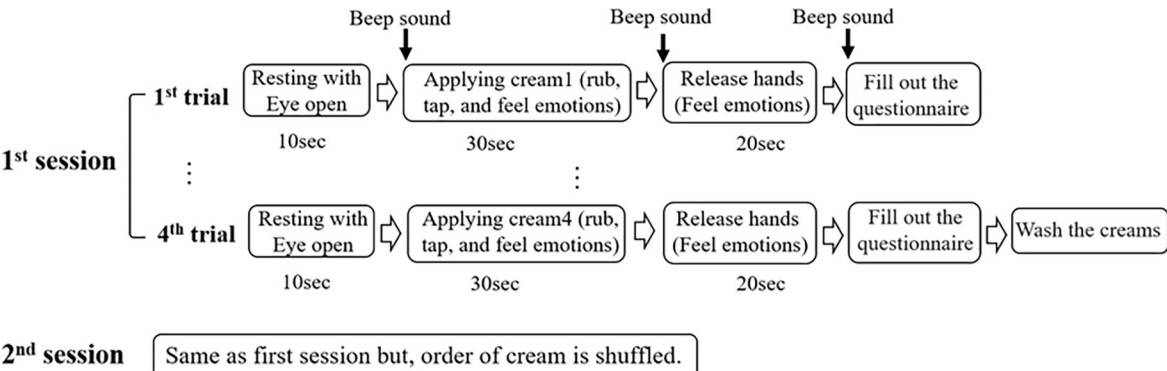

**Fig 2. Experimental procedure.**

sound is presented, the subject completed a satisfaction evaluation questionnaire. While the subject completed the questionnaire, the EEG equipment remained worn, and the subject was asked to keep the head and body movement as least as possible. In the second trial, a second cream was presented and measured in the same manner as in the first trial. Similarly, in the third and fourth trials, two creams were provided each. After applying the four creams, the subject washed them with water and proceeded with the second session. Four creams were applied following the same procedure as in the first session, and the order was presented at random, as in the first session. The cream order was randomized in all sessions for all subjects.

All creams used in the experiment were developed by Amorepacific. To remove the influence of emotion from the sense of smell, a fragrance was not added.

## 2–4. Questionnaire

The participants completed a questionnaire regarding their preference for the cream used immediately after applying each one. In the questionnaire, participants were asked to check the satisfaction level of the provided cream on a 9-point scale (1: very dissatisfied, 5: neutral, 9: very satisfied).

## 3. Data analysis

The overall data analysis procedure is shown in Fig 3, and the data preprocessing is described in Table 2.

Sixteen subjects were analyzed, excluding three subjects who had only one class label. The details of the preprocessing step are described in Sections 3–1 to 3–4.

### 3–1. Time-frequency analysis

Time-frequency analysis of EEG data was performed using Morlet wavelet decomposition. Morlet wavelets have a Gaussian shape defined by the spectral bandwidth ($\sigma f = 2f/C$) and wavelet duration ($\sigma t = 1/2\pi\sigma f$), where f is the center frequency. Here, C represents the number of wavelet cycles (or wavelet width), and the number of cycles at each frequency is the same. In our study, the specific parameters were $C = 14$, time resolution = 0.05 s, and frequency resolution = 1 Hz.

### 3–2. Windowing and reshaping

Fig 4 shows the spectrogram for one trial with one channel obtained after time-frequency analysis. Data are split into fixed length (3 s) windows with an overlap of 2 s (Fig 4(A)). We only

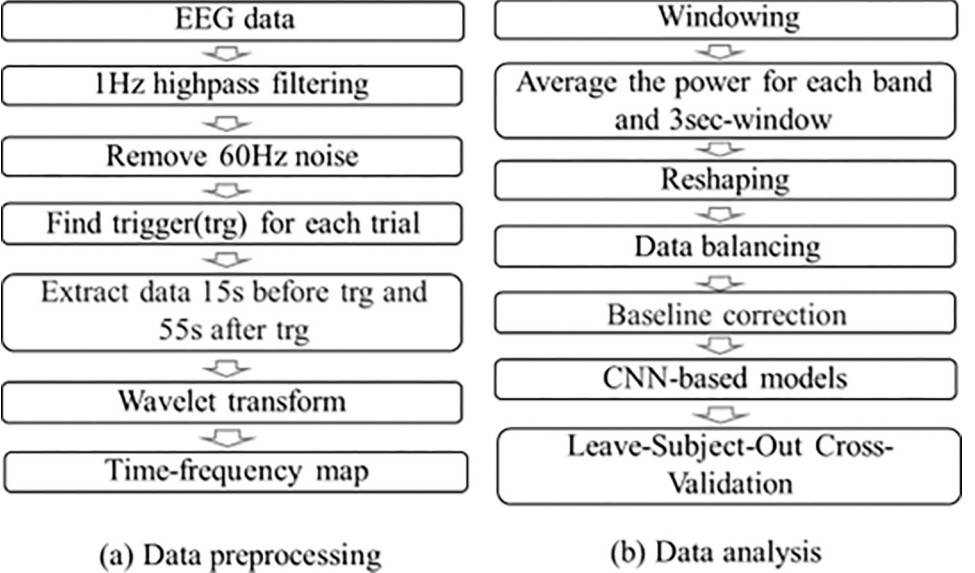

**Fig 3. Data analysis procedure.**

used the data for the 30 s while the applying creams for analysis. Because noise was found in many subjects with data for 20 s after releasing the hand, it was excluded from the analysis. A total of 28 windows were generated when data were collected while sliding in a 3 s window for 30 s.

For feature extraction, the 3 s spectrogram was divided into 4 frequency bands, and the average value was obtained for each frequency band (Fig 4(B)). Here, the four frequency bands are as follows: alpha (7–15 Hz), beta (15–33 Hz), low gamma (33–70 Hz), high gamma (70–120 Hz).

Then, the spectrogram was reshaped as in Fig 5. Fig 5(A) shows a 19-channel EEG layout. Fig 5(B) shows a matrix (size = 5×5) similar to the EEG layout. The empty space is filled with the average value of the surrounding channels. That is, ① is the average value of Fp1, F7, and

**Table 2. Data preprocessing.**

| 60 Hz noise remove | | Band stop filter(58 Hz-62 Hz) |
| --- | --- | --- |
| Extract the data | | -15 s– 55 s (trigger at 0 s) |
| Time-Frequency analysis | | Wavelet (Morlet) |
| TF parameter | Frequency resolution | 1:1:120 Hz |
| | Time resolution | 0.05 s |
| | Wavelet cycle | 14 |
| Windowing length | | 3 s |
| Percentage of overlap for sliding windows | | 66.7% |
| Extract features | | Four-frequency band (alpha, beta, low gamma, high gamma) |
| Reshaping | | Four-band merged matrix with EEG layout considered |
| Data balancing | | Using data augmentation |
| Baseline correction | | $f' = \frac{(f - \overline{f_{neg}})}{(\overline{f_{pos}} - \overline{f_{neg}})}$ |
| Remove subject | Only one class | Subj 5, 16, 19 |
| Number of subjects for analysis | | 16 |

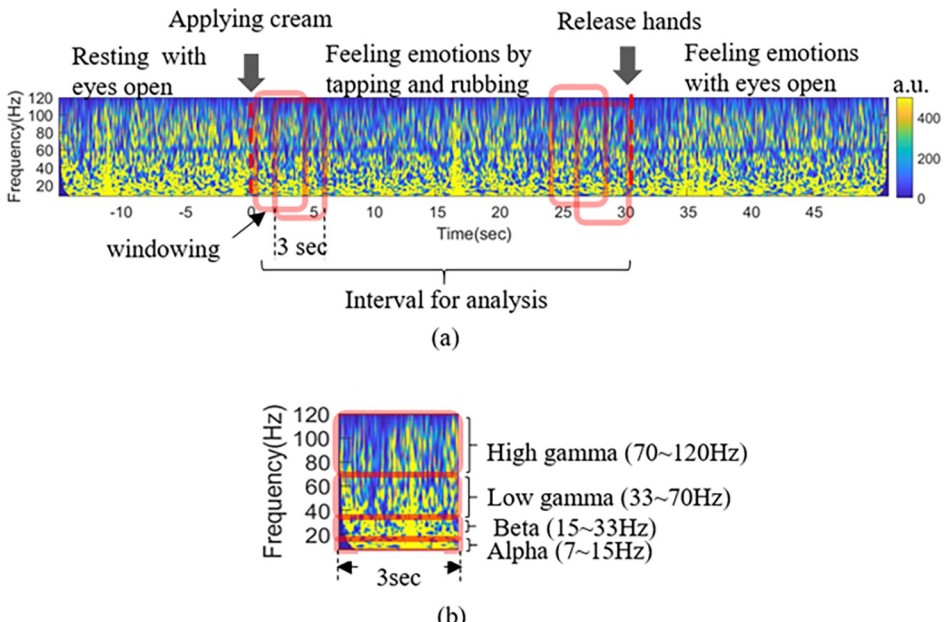

**Fig 4. Windowing with 2 s overlapping on the spectrogram.** (a) Spectrogram for one trial and 3s windowing (red box) with 2 s overlapping from 0 to 30 s. (b) 3 s window spectrogram and representation of each frequency band.

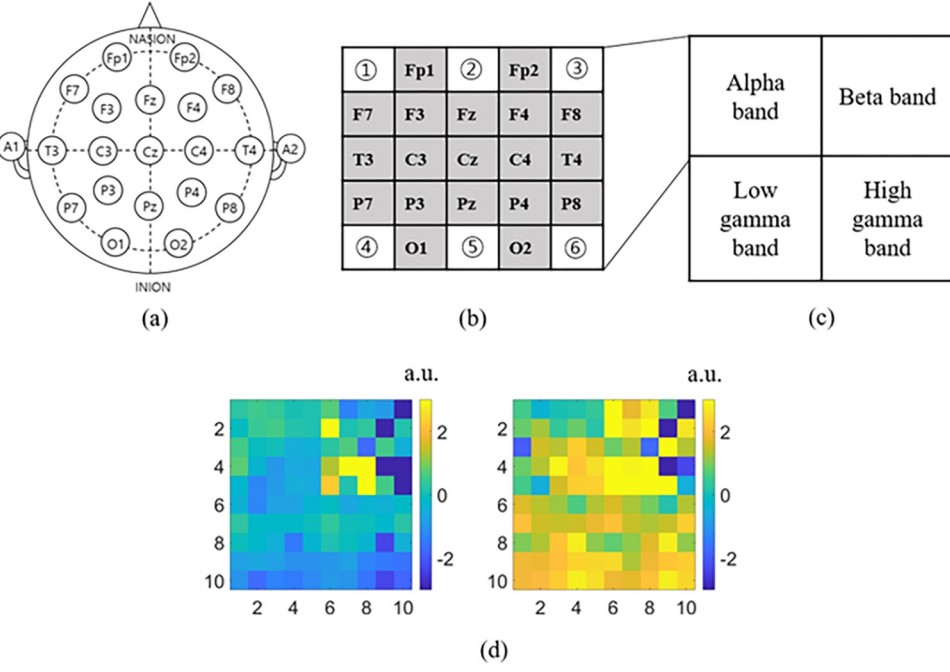

**Fig 5. Reshaping.** A merged four-band matrix considering the EEG channel position was used as input data. (a) 19-channel EEG layout. (b) Matrix (size = 5×5) considering EEG layout. (c) Merged matrix (size = 10×10) for four frequency bands. (d) An example of merged band images for subject 01's positive emotion (left) and negative emotion (right).

F3, ② is the average value of Fp1, Fz, and Fp2, ③ is the average value of Fp2, F4, and F8, ④ is the average value of P7, P3, and O1, ⑤ is the average value of Pz, O1, O2, ⑥ is filled with the average value of P8, P4, and O2. After that, a matrix is merged using four frequency bands (Fig 5(C)). Fig 5(D) shows two input images for one positive emotion (left) and one negative emotion (right) from subject 01.

## 3–3. Data balancing

For each subject, data imbalance occurs between the two classes. For example, in the case of subject 01, there are 140 and 56 positive and negative data, respectively. This is because five trials correspond to positive emotions, and two trials correspond to negative emotions. For all subjects, there are 1512 and 1008 data for positive and negative emotions, respectively. Ding et al. performed random over-sampling on EEG imblanced data in emotion recognition [59]. Lu et al. reported higher accuracy in random over-sampling than other method, random under-sampling, and cluster centroid [60]. Data augmentation was performed to balance the data. That is, after randomly selecting data from a small number of classes for each subject, increasing the data and making the number of data equal between the two classes. After data balancing, a total of 3360 data were made for both classes.

## 3–4. Baseline correction

Baseline correction was performed for each subject. Each subject has a different channel state. Because the measurement range is different for channels and subjects, baseline correction is required to obtain common spatial information between subjects. Baseline correction is performed for each channel and each band by subtracting the mean PSD of the negative class and dividing the difference between the positive mean PSD and negative mean PSD as shown in Eq 1.

$$f' = \frac{(f - \overline{f_{neg}})}{(\overline{f_{pos}} - \overline{f_{neg}})} \tag{1}$$

## 4. Classifier

We developed a deep learning-based classifier that classifies between positive and negative emotions using preprocessed data. An artificial neural network (ANN) is a statistical learning algorithm inspired by neural networks in biology (particularly the brain in the central nervous system of animals) in the fields of machine learning. An artificial neural network refers to an overall model that has problem-solving ability by learning artificial neurons (nodes) that form networks through synaptic bonds and changing the strength of synaptic bonds [61]. In general, the more complex the network, the more features are extracted and the better the classification result. However, in general, the more complex the network, i.e., the more parameters to train, the more training set samples are required. Therefore, high complexity model cannot be blindly pursued when designing neural network, and the network structure and number of sample sets must be balanced [62]. In general, in the training process, the learning curves of the training set and validation set are used to determine whether the network is overfitting or underfitting. The deep learning structures and algorithms used in this study are introduced in Sections 4–1 to 4–3.

## 4–1. Convolutional Neural Networks

In deep learning, a CNN is a type of ANN and is most commonly applied to image analysis. A CNN has the following characteristics [63,64]: 1) Maintains the shape of input/output data of each layer. 2) Effectively recognizes features with adjacent images while maintaining spatial

information of the image. 3) Feature extraction and learning of images with multiple filters. 4) A Pooling layer that collects and enhances the features of the extracted image. 5) Because the filter is used as a shared parameter, there are very few learning parameters compared to a general ANN. A CNN architecture includes several building blocks, such as convolution layers, pooling layers, and fully connected layers. A typical architecture consists of an iterative stack of multiple convolutions and pooling layers, followed by one or more fully connected layers [65]. The convolution step is a key element that extracts features, including the activation function, after applying a filter to the input data. In the pooling step, the image size can be greatly reduced when it is connected to the next convolutional layer or fully connected layer by summarizing the neighboring information (e.g., max pooling and average pooling). As it has been found useful to use CNNs to utilize such spatial information, they can be used to determine the correlation between physically adjacent channels by converting an EEG sequence into a 2D or 3D frame. In our study, emotion classification was conducted through CNN-based learning with 2D images as input.

## 4–2. Proposed CNN-based models

In this study, seven CNN-based models were implemented: 1) inception-like CNN with four-band merged input, 2) stacked CNN with four-band merged input, 3) stacked CNN with four-band parallel input, and 4) - 7) stacked CNN with single-band input (alpha, beta, low gamma, and high gamma).

**1) Inception-like CNN with four-band merged input.** Traditional CNNs use only a single-size kernel at each layer. GoogleNet's inception module shows excellent performance in image classification by extracting features of multiple sizes at once using kernels of different sizes in one layer. We developed an inception-like CNN model consisting of three modules, including bottleneck layer, convolutional layers, Elu activation, dropout layers, and fully connected layers (Fig 6). Table 3 describes the structure and hyper-parameters used in this model. The proposed inception-like CNN structure is based on the structure proposed by Zhang et al. [66].

**2) Stacked CNN with four-band merged input.** A stacked CNN model was constructed as in Fig 7. Input images are generated by merging four frequency bands considering channel location, as shown in Fig 5(C). As shown in Fig 7, input images of size 10×10 are trained on a model consisting of five convolutional layers, a dropout layer, and a fully connected layer consisting of the last two neurons. The filter (or kernel) size of the first and second convolutional layers is 2×2, and the filter size of the third to fifth layers is 3×3. The number of kernels in each convolutional layer is the same at 10, and the activation function is the Elu function. A dropout layer is added to the last convolution layer, and a softmax function is used for the output. Detailed model hyperparameters are listed in Table 4.

**3) Stacked CNN with four-band parallel input.** The parallel input CNN structure with four bands as input is shown in Fig 8. A matrix of size 5×5 is fed as input and trained on a model with three convolutional layers, a dropout layer, and a fully connected layer consisting of the last two neurons. The detailed model hyperparameters are listed in Table 4.

**4) - 7) Stacked CNN with single-band input (alpha, beta, low gamma, and high gamma).** We created a CNN model with a single-band as input. As shown in Fig 5(B), input images with size 5×5 for each of four bands are trained on a model with three convolutional layers. Detailed model hyperparameters are listed in Table 4.

## 4–3. Training parameters of the network

Training options for all models were as listed in Table 5.

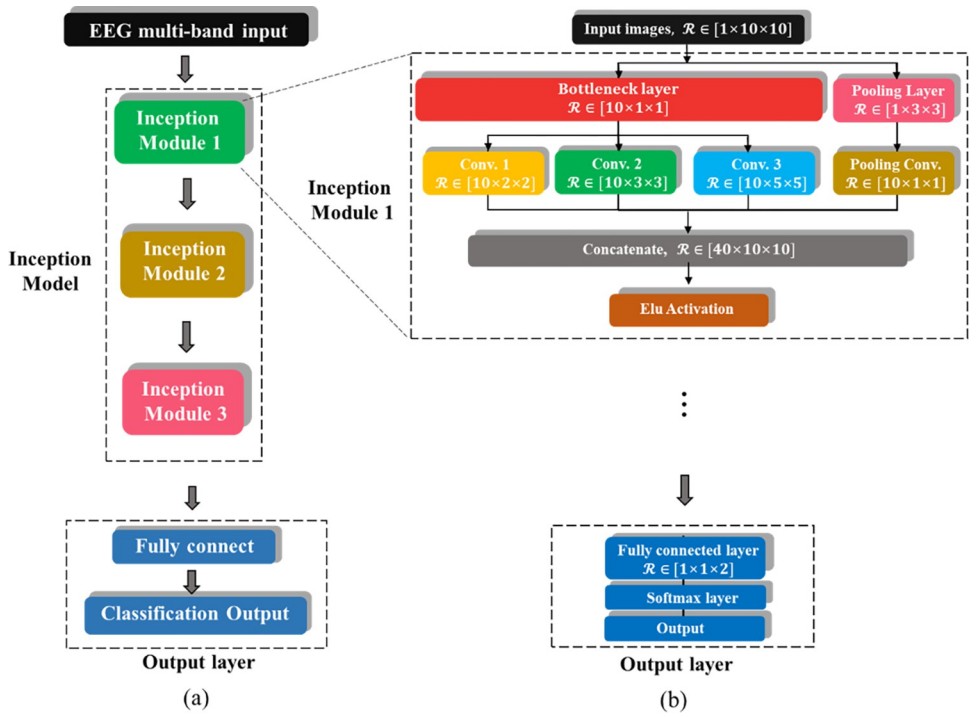

**Fig 6. Inception-like CNN model.**

**Table 3. Network structure and hyper-parameters for inception-like CNN Model.**

| Input size | | | 10 ×10 | |
|---|---|---|---|---|
| Normalization | | | none | |
| Network structure | Inception module 1 | Input size | 1×10×10 | |
| | | Bottleneck layer | Filter size = 10×1×1, padding: same | |
| | | Pooling layer | Filter size = 1×3×3, padding: same | |
| | | Convolution | Filter size = 10×2×2, 10×3×3, 10×5×5, padding: same | |
| | | Elu activation | | |
| | Inception module 2 | Input size | 40×10×10 | |
| | | Bottleneck layer | Filter size = 10×1×1, padding: same | |
| | | Pooling layer | Filter size = 1×3×3, padding: same | |
| | | Convolution | Filter size = 10×2×2, 10×3×3, 10×5×5, padding: same | |
| | | Elu activation | | |
| | Inception module 3 | Input size | 40×10×10 | |
| | | Bottleneck layer | Filter size = 10×1×1, padding: same | |
| | | Pooling layer | Filter size = 1×3×3, padding: same | |
| | | Convolution | Filter size = 10×2×2, 10×3×3, 10×5×5, padding: same | |
| | | Elu activation | | |
| | Output layer | Fully connected | Neuron, 2 | |
| | | Softmax | | |

The padding type 'same' means that the output feature map has the same spatial dimensions as the input feature map. Zero padding is introduced to ensure that the input map has the same shape as the output map on all sides.

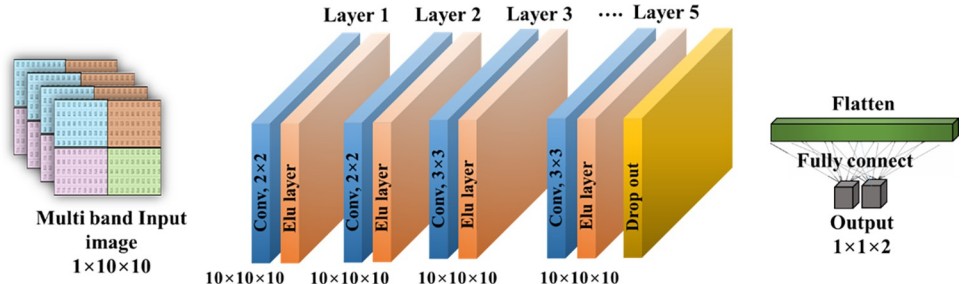

**Fig 7. A 5-layer CNN structure with merged four-band input.**

## 4–4. Filtering the network model output

According to Cimtay et al., a median filter was applied to the output data for emotion classification, assuming that the emotional state does not change over a short time interval (in the range of several seconds) in healthy individuals with good emotional regulation [51]. They applied a median filter of about 5 to 8 s to remove false alarms and increase the overall classification accuracy. In the SEED / DEAP / LUMED datasets, when the median filter was applied to two-class (Pos, Neg) classification, the one-subject-out evaluation result showed an average accuracy of 86.56% / 72.81% / 81.80%, and when not applied (82.94% / 68.60% / 77.11%), it increased by 3.62% / 4.21% / 4.69%, respectively. In our study, as applied by Cimtay et al., a 7 s median filter was applied to the output values.

## 4–5. Cross-validation and model performance evaluation

We performed LOSO CV for model evaluation. We strictly separated the training, validation, and test datasets, i.e, during each fold, we train our model on 14 subjects and validate on one subject, and test on the remaining one subject. Validation data are used to avoid overfitting during model training. Training network parameters (weights and biases) are saved at every

**Table 4. Network structure and hyper-parameters for three CNN-based models.**

|  |  | Single band input CNN | Multi-band input CNN | Parallel band input CNN | Hyperparameter |
|---|---|---|---|---|---|
| **Input size** |  | 1×5×5 | 1×10×10 | 1×5×5 |  |
| **Normalization** |  | none | none | none |  |
| **Network structure** |  | Convolution | Convolution | Convolution | 10×2×2, padding: same |
|  |  | Elu activation | Elu activation | Elu activation |  |
|  |  | Convolution | Convolution | Convolution | 10×2×2, padding: same |
|  |  | Elu activation | Elu activation | Elu activation |  |
|  |  | Convolution | Convolution | Convolution | 10×3×3, padding: same |
|  |  | Elu activation | Elu activation | Elu activation |  |
|  |  |  | Convolution |  | 10×3×3, padding: same |
|  |  |  | Elu activation |  |  |
|  |  |  | Convolution |  | 10×3×3, padding: same |
|  |  |  | Elu activation |  |  |
|  |  | Fully connected | Fully connected | Fully connected | Neuron, 2 |
|  |  | Softmax | Softmax | Softmax |  |

The padding type 'same' means that the output feature map has the same spatial dimensions as the input feature map. Zero padding is introduced to ensure that the input map has the same shape as the output map on all sides.

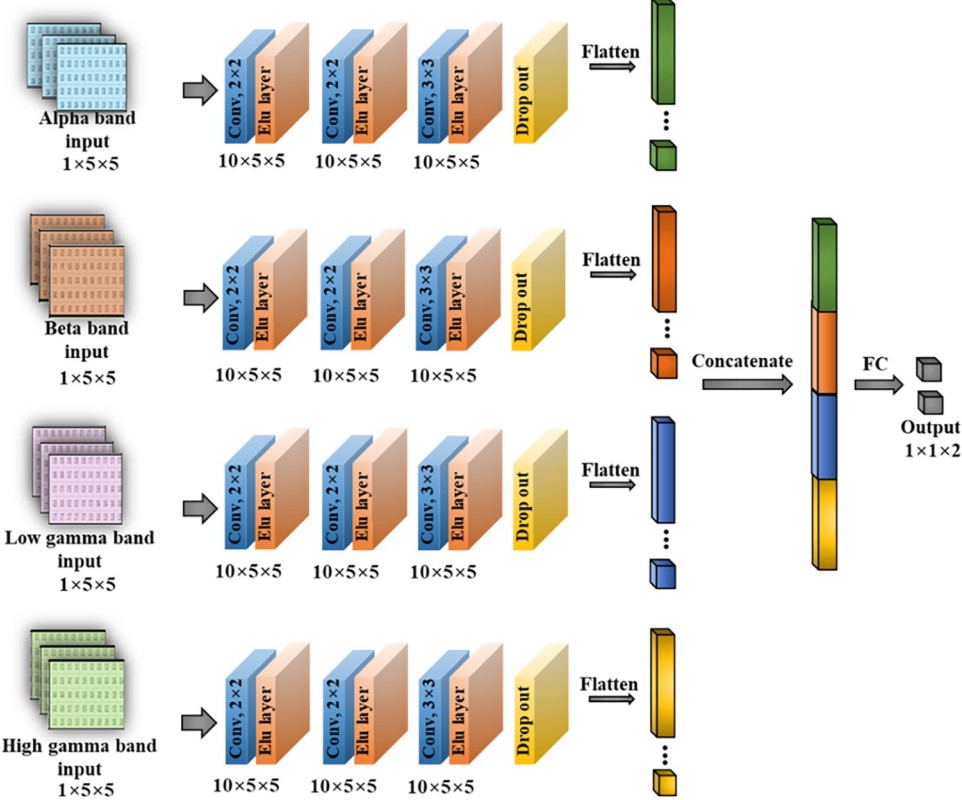

**Fig 8. A 3-layer CNN structure with four-band parallel input.**

epoch until the maximum epoch is reached. This is applied to the test data by using the network corresponding to the epoch when the accuracy of the verification data is the highest among the stored networks. Model performance uses accuracy, sensitivity, and specificity.

Sensitivity = TP / (TP+FN),

Specificity = TN / (TN+FP),

Accuracy = (TP+TN) / (TP+TN+FP+FN), where TP: True Positive, TN: True Negative, FP: False Positive, FN: False Negative       (2)

**Table 5. Training parameters of the network.**

| Property | Value |
| --- | --- |
| Mini-batch size | 128 |
| Weight / Bias initializer | Glorot Initialization / Zero Initialization |
| Loss function | Cross entropy |
| Validation frequency | Training size/mini-batch size |
| Optimizer | Adam |
| Learning rate | 0.0002 |
| Shuffle | Every epoch |
| Maximum epoch | 300 |
| Learning rate schedule | None |
| Hyperparameter search method | None |
| Environment | Win 10, 2 Parallel GPU(s), Matlab R2020b |
| Output classes | 2 classes (Positive, Negative) |

We use two subjects for validation and test, respectively. Fifteen training replicates are required in one test set. For example, if subject 01 is the test data and subject 02 is the valida-tion data, the remaining subjects 03 to 16 become the training dataset. Similarly, if subject 01 is the test data, the validation data may include one of subjects 03 to 16, and the remaining sub-jects become the training data. A total of 240 (16×15) training repetitions were performed for all subjects. Then, the average accuracy, average sensitivity, and average specificity for each subject were calculated. The performance evaluation of each model is obtained by taking the mean of the average values of all subjects.

In our study, we used compact-size images. We can reduce the image size by not repeating the information or padding the matrix with many zeros, as done by Chao et al. [46]. When the image size is small, the deep learning training is fast. Although the size of the image is small, EEG characteristics (frequency and spatial information) are well expressed.

In our study, the training speed was fast because of the small image size and simple model-ing. In addition, it is possible to perform fast calculations using MATLAB-based high-speed matrix operations and a GPU (GeForce RTX 2080 SUPER Gaming GT D6 8 GB, Santa Clara, CA, USA). For about 2000 images of size 10×10, the training time was about 2 min. When the LOSO CV was applied to all subjects, a total of 15×16 = 240 training sessions were required, and the required time was about 8 h.

## 5. Results and discussions

### 5–1. Score distributions of each cream

Fig 9 shows the distribution of scores for the four creams. Cream 1 is clustered near 1 point, and cream 2 has a wide distribution between 2 and 9 points. The other two are mainly distrib-uted in 7–9 points. For the preference scores (1 to 9 points) for the four creams, 1 to 4 points are labeled as negative, and 8 to 9 points are labeled as positive. A total of 1512 positive data and 1008 negative data are generated. Data is balanced between the two classes through data augmentation, and a total of 3360 data are made for both classes.

### 5–2. Baseline correction

Because the baseline is different for each subject, baseline correction is required for cross-sub-ject application. Fig 10 shows baseline correction applied to EEG data. For two band channels (beta band-C3, gamma1 band-C3) of three subjects, the data distributions corresponding to the positive and negative classes show before (top) and after (bottom) the baseline correction. Here, gamma1 is low gamma (33-70Hz). The reason for choosing the electrode and frequency band presented here is that differences between the two emotions were reported in beta and gamma bands near the left temporal region in existing emotion studies [67]. The three subjects were selected as subjects with high classification accuracy. When baseline correction is not per-formed, the data distribution of each subject shows a different distribution.

As mentioned by Lan et al, the baseline of EEG is different for each subject, and it can be seen that they have different domains from the perspective of data analysis [68]. Accuracy is very poor when running tests on completely new domains that have not included in the train-ing. Lan et al. applied domain adaptation to SEED datasets, and according to the experimental results, the accuracy could be significantly improved by 7.25%– 13.40% when using the domain adaptation technique compared to the reference in which the domain adaptation tech-nique was not used. However, they did not show a significant effect in the DEAP dataset (aver-age accuracy when applying domain adaptation technique = 48.93%). Fan et al. reported the same baseline correction method as ours [69]. They developed a two-step function correction method, baseline function subtraction, and individualized function normalization to prepare

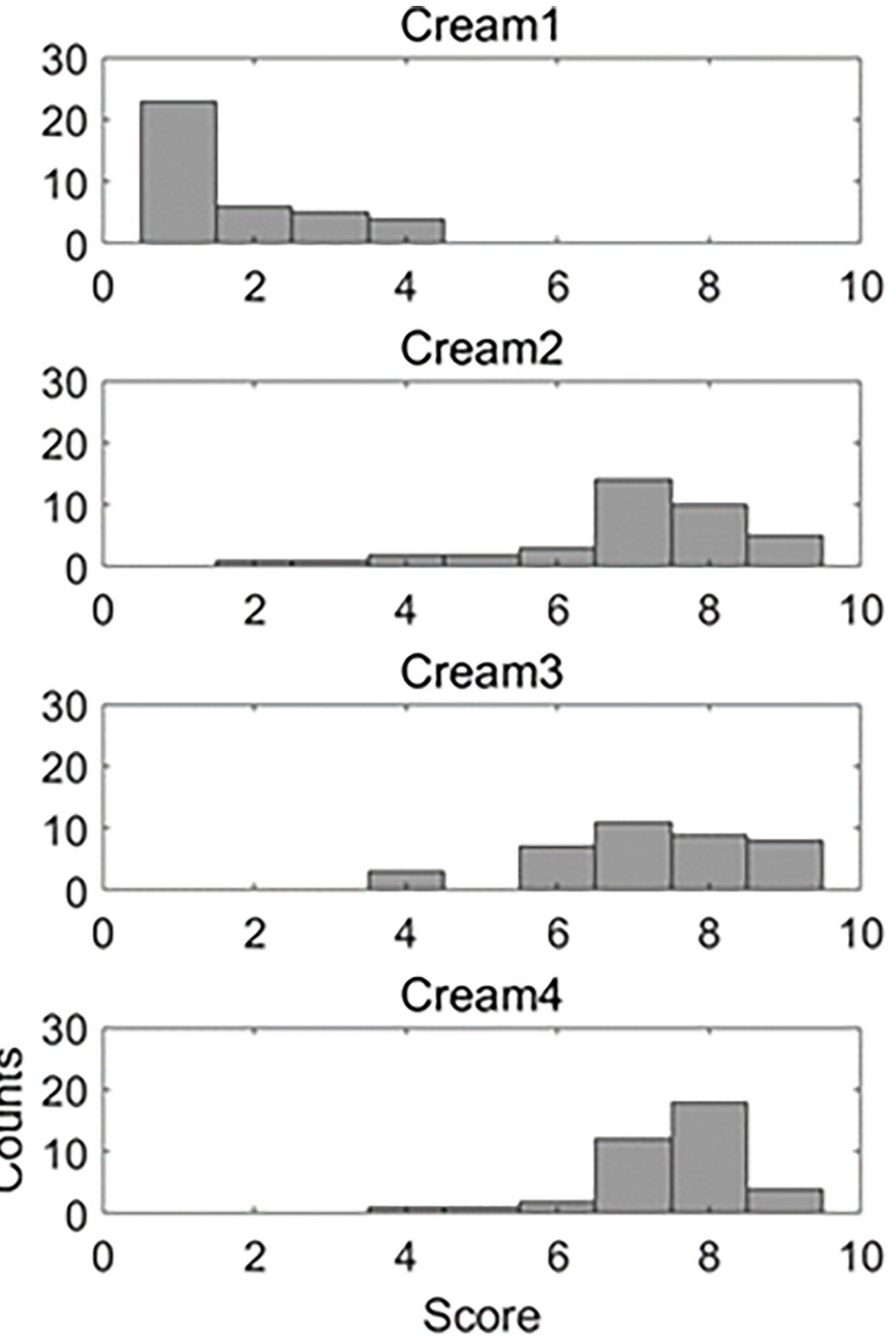

**Fig 9. Score distributions for four creams.**

EEG functions for group-level influence and workload recognition. The normalization step readjusted the feature range of each subject based on the average of features belonging to the low-intensity and high-intensity classes. As in our study, this requires each subject to have two classes of data (e.g., low engagement and high engagement).

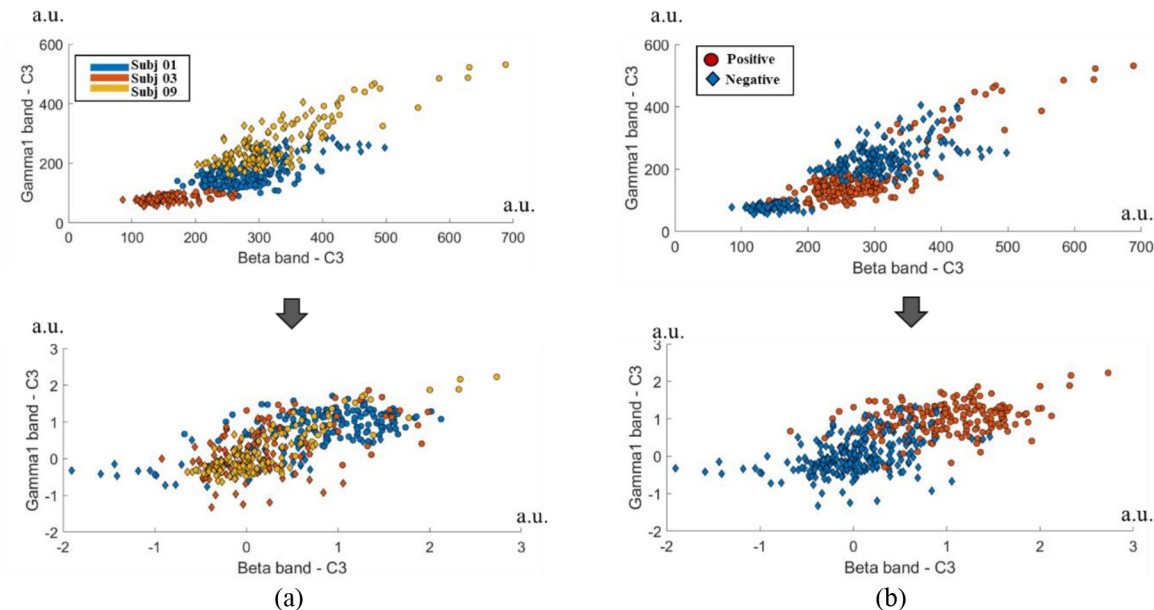

**Fig 10.** Comparison before (upper) and after (below) baseline correction for two band channels (beta band-C3, Gamma1 band-C3 of three subjects). (a) Two-class data distribution of three subjects, displayed in a different color for each subject regardless of class. (b) Two-class data distribution of three subjects. Each class is displayed in a different color regardless of the subject.

## 5–3. Performance of subject-independent models

Fig 11 shows the accuracy and loss graphs for the iterative learning time of training for the Stacked CNN, one of the proposed models. Validation accuracy and loss were calculated for each epoch during training. Here, the training dataset was subject 01 and subject 04 to subject 16, the validation dataset was subject 03, and the test dataset was subject 02. Maximum epoch was 300. We used validation data to avoid overfitting the model. Two methods (maximum validation accuracy and minimum validation loss) were used to find the best model. Accuracy was calculated using the test dataset for both models. Accuracy of test subject 02 was obtained at 75.0% and 71.4% for maximum validation accuracy and minimum validation loss, respectively.

Fig 12 shows the accuracies for seven classification models. The results were obtained based on maximum validation accuracy. Fig 13 shows the accuracies for seven classification models. The results were obtained based on minimum validation loss. Fig 14 shows the average accuracies, average sensitivities, and average specificities for 16 subjects for all models. In addition, the standard deviation is given for each result. The average test accuracies based on the maximum validation accuracy were 73.2%, 75.4%, 73.9%, 68.8%, 68.0%, 70.7%, and 69.7% for the inception-like CNN, stacked CNN, parallel band input CNN, alpha band CNN, beta band CNN, gamma1 band CNN, and gamma2 band CNN, respectively. The average test accuracies based on the minimum validation loss were 70.9%, 74.6%, 69.6%, 67.8%, 67.2%, 69.9%, and 69.6%, respectively.

For all models, the accuracy obtained based on the maximum validation accuracy was higher than that based on the minimum validation loss. We also found that, in the results based on both methods, the accuracy was higher than when only single band was used in the model trained using all four frequency bands. That is, the average accuracy of the three models, inception-like CNN, stacked CNN, and parallel input CNN, was 74.1% and 71.7% in the two validation data-based models, respectively. The average accuracy for each band model was

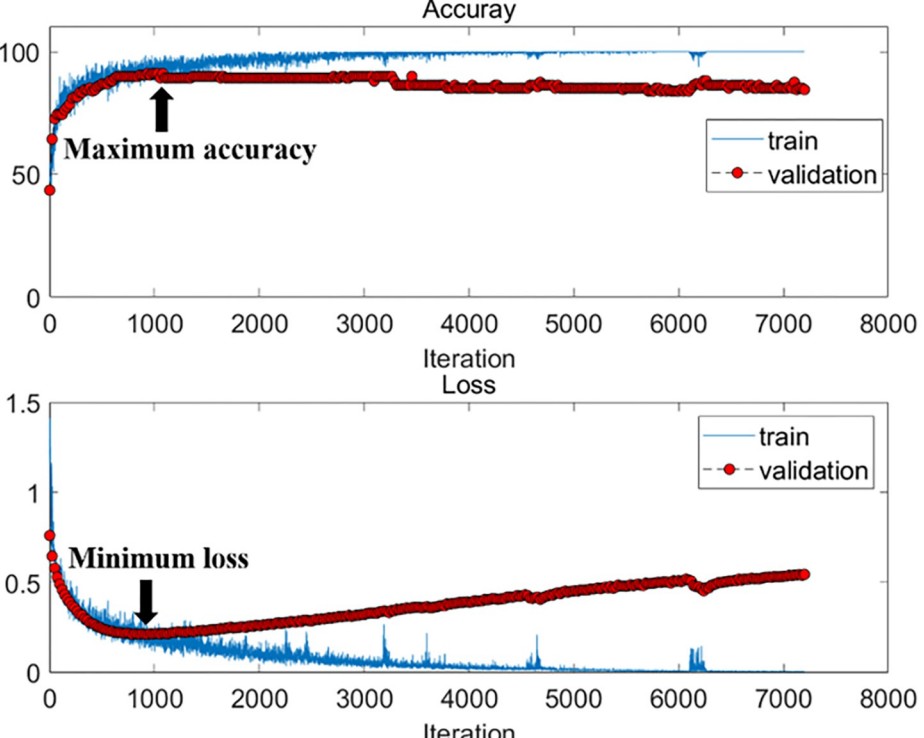

**Fig 11.** Accuracy (top) and loss (bottom) against training time in iterations for train dataset (subject 01, subject 04 -subject 16), validation dataset (subject 03), and test dataset (subject 02). Maximum epoch is 300. Two methods (maximum accuracy and minimum loss) were used to find the best models.

69.3% and 68.6% for the two validation data-based models, respectively. That is, when all bands were used, the averages of the two validation data-based models were 4.8% and 3.1% higher than when only each band was used. The stacked CNN showed the highest accuracy (75.4%, 74.6%) in both methods (based on validation data). Also, in each band, both methods (based on validation data) showed the highest accuracy (70.7%, 69.9%) in the low gamma band. This is the same as the result of good emotion classification in the gamma band in other literature [70–76]. In Wang et al.'s paper, they obtained very large activation maps in the high-frequency region of about 25–60 Hz in images corresponding to positive, negative, and neutral emotions [76].

The sensitivity was higher than specificity in all models. This means that the model classifies positive classes better than negative classes.

In the future, we plan to conduct research on identifying features that distinguish emotions (EEG channel and frequency band) using Grad-CAM [77] or Layer-Wise Relevance Propagation [78].

## 6. Conclusion

We developed CNN-based models using EEG for emotional recognition while applying creams with four different textures. In our study, rather than learning EEG raw data as it is, we used a knowledge-based neural network model using well-known EEG frequency features. We extracted frequency features as four frequency bands (alpha, beta, low gamma, and high gamma) for each channel in 19-channel EEG, a matrix was created considering the channel position, and the four-band matrix was merged.

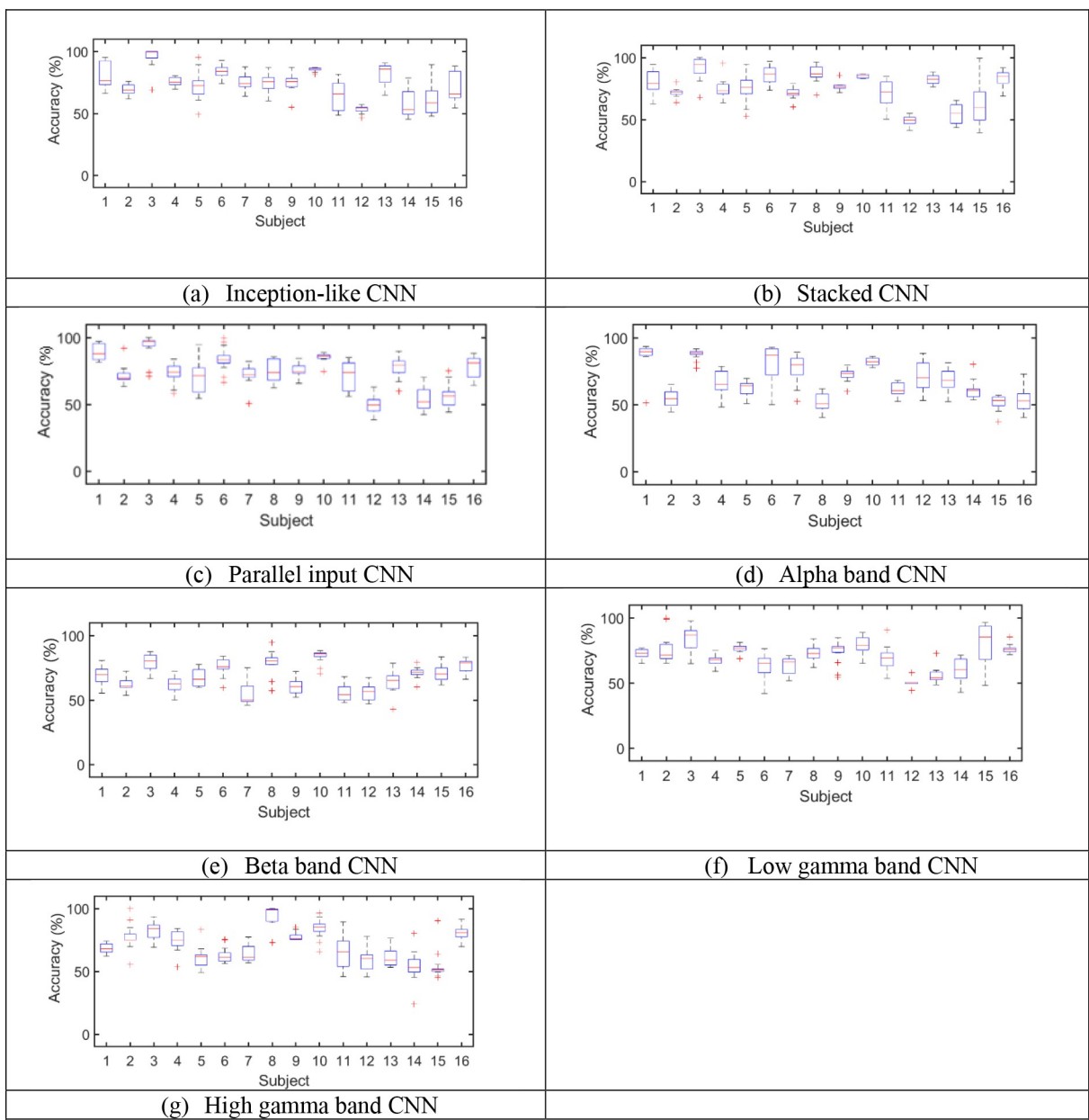

(a) Inception-like CNN
(b) Stacked CNN
(c) Parallel input CNN
(d) Alpha band CNN
(e) Beta band CNN
(f) Low gamma band CNN
(g) High gamma band CNN

**Fig 12. Maximum Accuracies for seven CNN-based models based on validation accuracy.**

The average accuracies of all subjects based on the maximum validation accuracy were 73.2%, 75.4%, 73.9%, 68.8%, 68.0%, 70.7%, and 69.7% for inception-like CNN, stacked CNN, parallel band input CNN, alpha band CNN, beta band CNN, low gamma band CNN, and high gamma band CNN, respectively. The average accuracies of all subjects based on the minimum validation loss were slightly lower for all models than those based on the maximum validation accuracy. The best performance was found to be 75.4% for the stacked CNN based on maximum validation accuracy.

It is noteworthy that the model performance in our study was evaluated by thoroughly distinguishing the train, validation, and test dataset. Research was conducted to develop a general-purpose model applicable to cross-subjects. Although it is a difficult task to find a

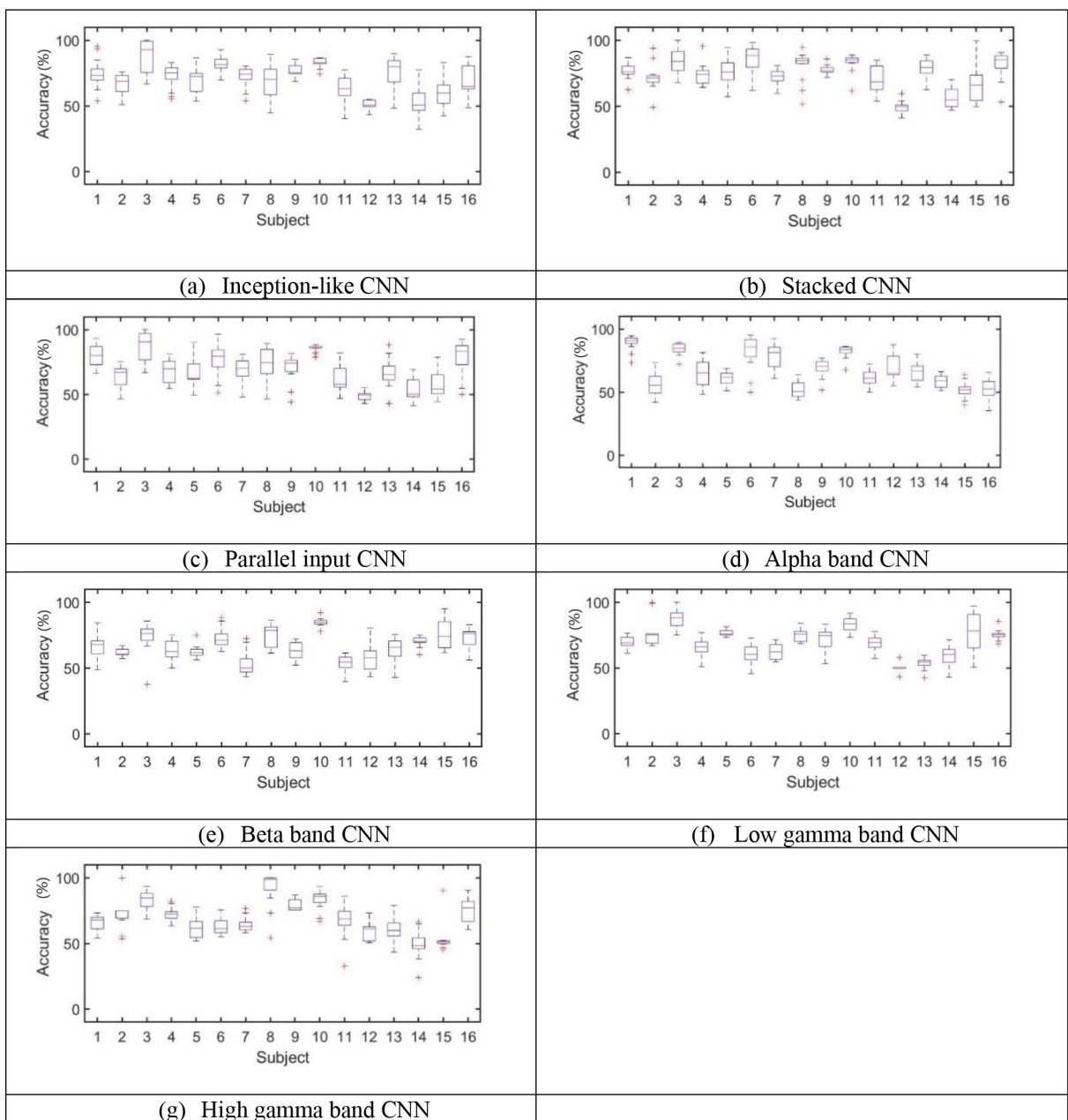

**Fig 13. Accuracies for seven CNN-based models based on minimum validation loss.**

classification model by dividing the test and validation for a small number of subjects (n = 16), we confirmed the possibility of generating a subject-independent model to classify the emotions evoked by cosmetic creams and obtained very remarkable results.

In our study, a median filter with a length of 7 s was used to lower the false alarms in the output value. We obtained an accuracy of 72.0% for the maximum validation accuracy for the merged-band CNN model without the median filter. After median filtering, we achieved 75.4% for the merged-band CNN model. That is, there was an increase of about 3.4%.

Although we analyzed small number of subjects, this study confirmed the possibility of an emotion (like/dislike) classification using EEG. We plan to conduct experiments on more

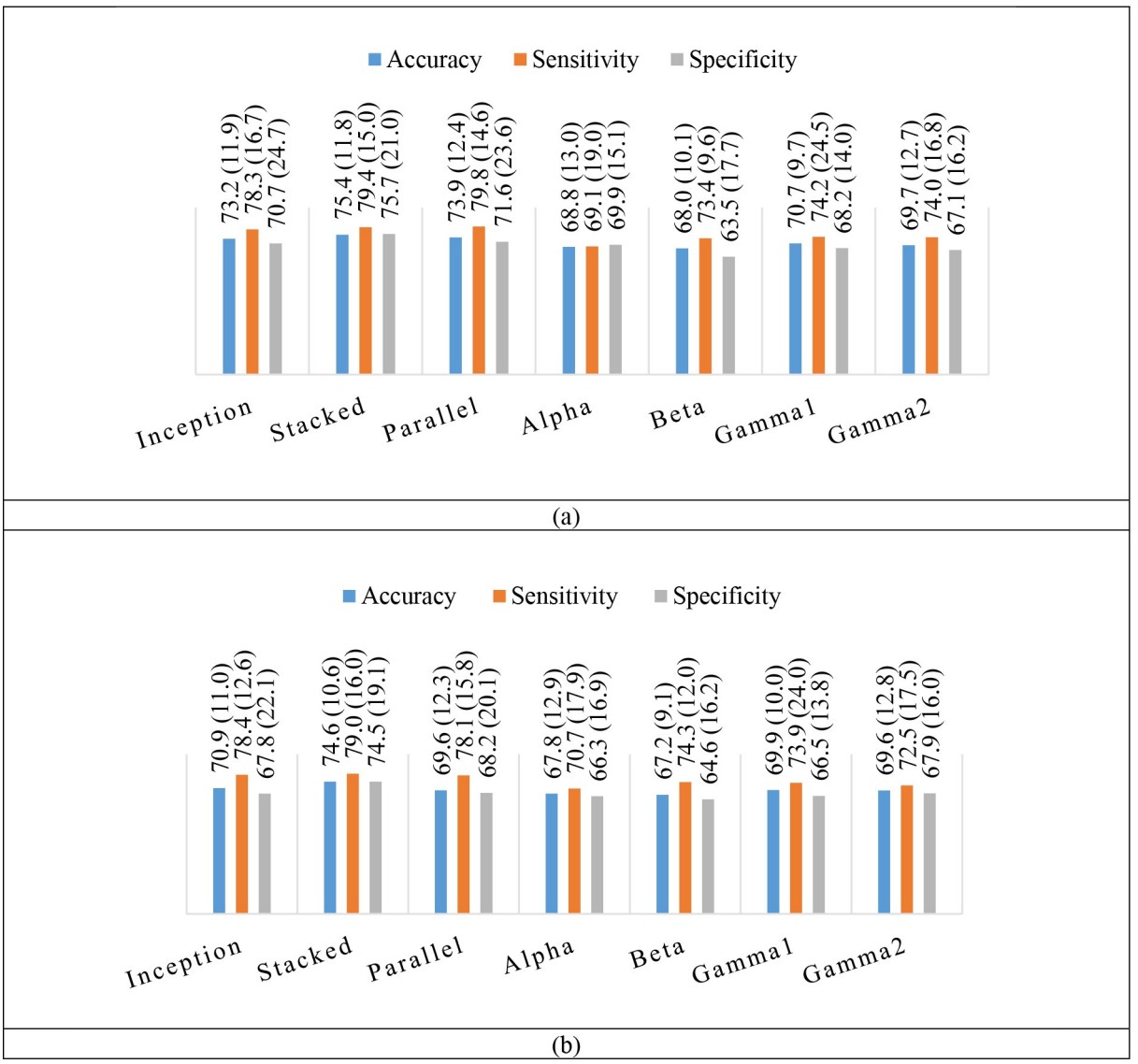

**Fig 14. Average classification results of 7 CNN-based models for positive and negative emotions in subject-independent models.** The standard deviation is shown in parentheses next to each result. (a) Test results based on maximum validation accuracy. (b) Test results based on minimum validation loss.

subjects in the future, and we expect that future results can provide more feasible implication to evaluate objective emotional benefits from cosmetics use through EEG.

## Author Contributions

**Conceptualization:** Jieun Kim, Dong-Uk Hwang, Edwin J. Son, Sang Hoon Oh, Youngkyung Kim, Gusang Kwon.

**Data curation:** Gusang Kwon.

**Methodology:** Jieun Kim, Edwin J. Son, Whansun Kim.

**Project administration:** Youngkyung Kim.

**Resources:** Youngkyung Kim, Gusang Kwon.

**Software:** Jieun Kim, Whansun Kim.

**Validation:** Dong-Uk Hwang, Edwin J. Son, Whansun Kim.

**Writing – original draft:** Jieun Kim.

**Writing – review & editing:** Dong-Uk Hwang, Edwin J. Son, Sang Hoon Oh, Youngkyung Kim, Gusang Kwon.

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
