## [Decision Letter · Decision Letter 0]

23 May 2022

PONE-D-22-07522Emotion recognition while applying cosmetic cream using deep learning from EEG data; cross-subject analysisPLOS ONE

Dear Dr. Kim,

Thank you for submitting your manuscript to PLOS ONE. After careful consideration, we feel that it has merit but does not fully meet PLOS ONE’s publication criteria as it currently stands. Therefore, we invite you to submit a revised version of the manuscript that addresses the points raised during the review process.

We look forward to receiving your revised manuscript.

Kind regards,

Anwar P.P. Abdul Majeed

Academic Editor

PLOS ONE

Journal Requirements:

Reviewers' comments:

Reviewer's Responses to Questions

**Comments to the Author**

1. Is the manuscript technically sound, and do the data support the conclusions?

Reviewer #1: Yes

Reviewer #2: Yes

2. Has the statistical analysis been performed appropriately and rigorously? 

Reviewer #1: Yes

Reviewer #2: Yes

3. Have the authors made all data underlying the findings in their manuscript fully available?

Reviewer #1: No

Reviewer #2: No

4. Is the manuscript presented in an intelligible fashion and written in standard English?

Reviewer #1: No

Reviewer #2: No

5. Review Comments to the Author

Reviewer #1: In this paper, the authors have proposed EEG based emotion recognition while applying cosmetic cream using deep learning. The work is interesting; however, it needs minor revisions in accordance with the following comments-

The introduction of this manuscript is very long. It must be concise. I am going to suggest how authors may shorten the introduction.

-First three paragraphs (1st paragraph, section 1.1, section 1.2) could be merged into one paragraph. Last sentence of first paragraph (In this study, meaningful results) should be cited properly.

- Section 1.3, 1.4, 1.5 contain very basic information. I think no need a detailed explanation of this information. Authors may use a small paragraph to present this information with proper citation.

-In section 1.6 and 1.7, the authors reviewed lots of related studies very comprehensively. These sections should also be concise.

-In the last paragraph, the authors should mention some research gaps based on the related study above. Then, the authors should mention their contributions to overcome those gaps. These contributions must be very specific, informative and not more than 4 points. Here, please avoid generalizations like- point 6 of section 1-8: “developing CNN models with various structures”.

-In section 3-2 (Data balancing), the authors performed data augmentation to balance the data. To explain the process, the authors made this statement- “That is, after randomly selecting data from a small number of classes for each subject, increasing the data and making the number of data equal between the two classes.” Is there any citation to support this statement? In the case of a random selection of sample/observation, if the majority of wrong sample/observations are selected, then the accuracy will be lower than actual. Conversely, if the majority of right sample/observations are selected, then the accuracy will be higher than the actual. The authors should clarify this issue.

-In Figure 5, the authors merged time-frequency image of 19 electrodes in 5*5 matrix. That’s a great idea. However, if the number of the electrodes is different i.e., 5/8/30/60/64 or more, how the image will be merged? The authors should clarify this issue.

-In the result and discussion, the author claimed that stacked CNN achieved the highest accuracy over other methods. It will be excellent if authors explain- for which properties of stacked CNN, it achieved the highest accuracy? Why other methods achieved comparatively lower accuracy? It is normal that the accuracy of stacked CNN is also very low as compared to the other EEG based application or EEG based emotion recognition. However, the authors should explain the reasons behind this lower accuracy.

-In pages 17 and 18, the authors mentioned the padding in the last column of this table. I suggest mentioning the number of padding for this experiment.

-In page 19-20 (Table 5), the authors should clarify why they used a specific Learning rate of 0.0002 and a maximum epoch of 300?

-The caption for Figure 4, 5, 11, 12, 13, and 14 should be concise.

-Please improve the presentation of Figure 3, Figure 9 (just remove free space), Figure 14

-English should be revised in any section of the paper because some typos are present throughout the manuscript and the used English should be polished by further proofreading.

Reviewer #2: The study aims to investigated different CNN architecture towards classifying cosmetic preferences. The following are my comments:

Introduction:

Quite lengthy, some parts could be merged and summarized. Consider trimming down the content and highlight the major findings of previous studies before describing the aim of the present investigation and how it addresses the gap in the body of knowledge

Methods:

Data Balancing: Augmentation - was the selected data (small number of classes) replicated? It was not specifically mentioned how it was increased.

4.2.1 - Why only the inception model employed, since there are other transfer learning models available, i.e., VGG16 amongst others?

Table 4, how was the hyperparameters selected? was it optimised or is it merely heuristic?

Results: Table 12 & 13, missing (%) symbol on the Y-axis.

6. PLOS authors have the option to publish the peer review history of their article (what does this mean?). If published, this will include your full peer review and any attached files.

Reviewer #1: No

Reviewer #2: No

---

## [Author Response · Author response to Decision Letter 0]

20 Jul 2022

I have completed responding to the reviewer's comments in the Response to "Reviewers.docx file".

---

## [Decision Letter · Decision Letter 1]

24 Aug 2022

Emotion recognition while applying cosmetic cream using deep learning from EEG data; cross-subject analysis

PONE-D-22-07522R1

Dear Dr. Kim,

We’re pleased to inform you that your manuscript has been judged scientifically suitable for publication and will be formally accepted for publication once it meets all outstanding technical requirements.

Kind regards,

Anwar P.P. Abdul Majeed

Academic Editor

PLOS ONE

Additional Editor Comments (optional):

Reviewers' comments:

Reviewer's Responses to Questions

**Comments to the Author**

1. If the authors have adequately addressed your comments raised in a previous round of review and you feel that this manuscript is now acceptable for publication, you may indicate that here to bypass the “Comments to the Author” section, enter your conflict of interest statement in the “Confidential to Editor” section, and submit your "Accept" recommendation.

Reviewer #1: All comments have been addressed

2. Is the manuscript technically sound, and do the data support the conclusions?

Reviewer #1: Yes

3. Has the statistical analysis been performed appropriately and rigorously? 

Reviewer #1: Yes

4. Have the authors made all data underlying the findings in their manuscript fully available?

Reviewer #1: Yes

5. Is the manuscript presented in an intelligible fashion and written in standard English?

Reviewer #1: Yes

6. Review Comments to the Author

Reviewer #1: I have read the manuscript entitled “Emotion recognition while applying cosmetic cream using deep learning from EEG data; cross-subject analysis”. The revisions that authors made to the manuscript are very effective in addressing all the concerns. My decision is to accept the manuscript for publication

in PLOS ONE.

7. PLOS authors have the option to publish the peer review history of their article (what does this mean?). If published, this will include your full peer review and any attached files.

Reviewer #1: **Yes: **Mamunur Rashid

---

## [Editor Report · Acceptance letter]

13 Oct 2022

PONE-D-22-07522R1 

Emotion recognition while applying cosmetic cream using deep learning from EEG data; cross-subject analysis 

Dear Dr. Kim:

I'm pleased to inform you that your manuscript has been deemed suitable for publication in PLOS ONE. Congratulations! Your manuscript is now with our production department. 

Kind regards, 

on behalf of

Dr. Anwar P.P. Abdul Majeed 

Academic Editor

PLOS ONE